# Effectiveness of and Inequalities in COVID-19 Epidemic Control Strategies in Hungary: A Nationwide Cross-Sectional Study

**DOI:** 10.3390/healthcare11091220

**Published:** 2023-04-25

**Authors:** Rahul Naresh Wasnik, Ferenc Vincze, Anett Földvári, Anita Pálinkás, János Sándor

**Affiliations:** 1Department of Public Health and Epidemiology, Faculty of Medicine, University of Debrecen, H-4002 Debrecen, Hungary; rahul.naresh@med.unideb.hu (R.N.W.); vincze.ferenc@med.unideb.hu (F.V.); 2Doctoral School of Health Sciences, University of Debrecen, H-4002 Debrecen, Hungary; anettefoldvari@gmail.com; 3ELKH-DE Public Health Research Group, Department of Public Health and Epidemiology, Faculty of Medicine, University of Debrecen, H-4002 Debrecen, Hungary; palinkas.anita@med.unideb.hu

**Keywords:** COVID-19 epidemic, epidemiological measures, effectiveness, social status, inequalities

## Abstract

Introduction: Before the mass vaccination, epidemiological control measures were the only means of containing the COVID-19 epidemic. Their effectiveness determined the consequences of the COVID-19 epidemic. Our study evaluated the impact of sociodemographic, lifestyle, and clinical factors on patient-reported epidemiological control measures. Methods: A nationwide representative sample of 1008 randomly selected adults were interviewed in person between 15 March and 30 May 2021. The prevalence of test-confirmed SARS-CoV-2 infection was 12.1%, of testing was 33.7%, and of contact tracing among test-confirmed infected subjects was 67.9%. The vaccination coverage was 52.4%. Results: According to the multivariable logistic regression models, the occurrence of infection was not influenced by sociodemographic and lifestyle factors or by the presence of chronic disease. Testing was more frequent among middle-aged adults (aOR = 1.53, 95% CI 1.10–2.13) and employed adults (aOR = 2.06, 95% CI 1.42–3.00), and was more frequent among adults with a higher education (aOR_secondary_ = 1.93, 95% CI 1.20–3.13; aOR_tertiary_ = 3.19, 95% CI 1.81–5.63). Contact tracing was more frequently implemented among middle-aged (aOR41-7y = 3.33, 95% CI 1.17–9.45) and employed (aOR = 4.58, 95% CI 1.38–15.22), and those with chronic diseases (aOR = 5.92, 95% CI 1.56–22.47). Positive correlation was observed between age groups and vaccination frequency (aOR41-70y = 2.94, 95% CI 2.09–4.15; aOR71+y = 14.52, 95% CI 7.33–28.77). Higher than primary education (aOR_secondary_ = 1.69, 95% CI 1.08–2.63; aOR_tertiary_ = 4.36, 95% CI 2.46–7.73) and the presence of a chronic disease (aOR = 2.58, 95% CI 1.75–3.80) positively impacted vaccination. Regular smoking was inversely correlated with vaccination (aOR = 0.60; 95% CI 0.44–0.83). Conclusions: The survey indicated that testing, contact tracing, and vaccination were seriously influenced by socioeconomic position; less so by chronic disease prevalence and very minimally by lifestyle. The etiological role of socioeconomic inequalities in epidemic measure implementation likely generated socioeconomic inequality in COVID-19-related complication and death rates.

## 1. Introduction

On 30 January 2020, the World Health Organization (WHO) declared the SARS-CoV-2 outbreak a public health emergency of international concern [1,2]. As of 3 May 2021, 169,597,415 confirmed cases had been reported worldwide [3]. Even before the pandemic was declared, the prevalence of coronavirus disease 2019 (COVID-19), caused by SARS-CoV-2, had increased significantly in some European countries and placed a strain on healthcare systems [4]. The European Center for Disease Prevention and Control (ECDC) estimates that 32,430,146 cases and 721,375 deaths had been recorded in the European Union by 31 May 2021 [5]. Due to the restricted effectiveness of treatment for SARS-CoV-2 infection, before vaccines were developed and widely administered, epidemiological measures (sanitation, hygiene, testing, case isolation, contact tracing, and quarantine) were the only means of containing SARS-CoV-2 [6,7,8,9,10]. Therefore, the effectiveness of epidemiologic control measures determined the extent and consequences of the COVID-19 epidemic. “Which control measures have been more or less successful to mitigate the spread?” [11] and “Which population areas will sustain R_o_ (reproductive number) >1, and what are the determinants of this degree of transmission?” [12] were among the open questions formulated at the beginning of the pandemic. Until now, these kind of questions have remained unanswered, and need thorough investigation to draw conclusions from experiences of COVID-19 management.

The number of ECDC-reported COVID-19-related deaths in Hungary was 29,733 by the end of May 2021. While the relative risk of infection in Hungary (cumulative rate of infection: 8.2%) compared to the EU average (cumulative rate of infection: 7.2%) was 1.14; the relative risk for COVID-19-related death was 1.89 in Hungary. The relative risk of COVID-19-related death corrected for the infection rate was 1.66 in Hungary, and the number of expected deaths was 17,896 on the basis of the EU infection and death rates. Consequently, 11,837 deaths were attributable to condition in Hungary that deviated from the EU average which determined the effectiveness of epidemic control measures and the treatment of infected patients [13].

Detailed analysis of the excess mortality confirmed the remarkable excess deaths during the COVID-19 pandemic in Hungary. It was demonstrated that Hungary belonged to the tertile of the EU countries with the highest excess mortality rates [14].

According to ecological studies, the socioeconomic position influenced the epidemic in Hungary. The settlement level deprivation showed significant inverse association with infection rate and significant association with mortality rate [15]. There was a significant negative role of deprivation in the respect of vaccination coverage too [16].

As compliance with epidemiological measures has not yet been evaluated in Hungary, our study aimed (1) to describe patient experiences with COVID-19-related testing, contact tracing, and vaccination, (2) to evaluate the impact of sociodemographic, lifestyle, and clinical factors on patient-reported epidemiological measures, and (3) determine the contribution of non-perfect epidemiological control measures to the mortality excess.

## 2. Materials and Methods

### 2.1. Setting

This investigation was a population-based cross-sectional study. Data were obtained from the 2021 International Social Survey Program (ISSP) conducted from 15 March to 30 May 2021, during the third wave of COVID-19. The survey was completed with COVID-19 epidemic-related questions. Data from the survey database were provided for the analysis. A representative sample of 1008 Hungarian adults over 18 years of age were randomly selected from the country’s whole population (using the national registry of the Hungarian population as sampling frame) and interviewed in person by trained interviewers [17].

### 2.2. Outcome Variables

Each outcome measure was determined by the participants’ self-declaration. COVID-19 infection was ascertained if a participant had an infection with the SARS-CoV-2 virus confirmed by laboratory findings. Participants tested for SARS-CoV-2 infection under any circumstances before the time of the survey were distinguished from those who did not undergo testing. Patients with confirmed infection declared whether they were interviewed by health care professionals or epidemiologists about their contact persons. The implementation of contact tracing was registered accordingly. The mass vaccination was in progress at the time of the data collection, and partial vaccination was rare among Hungarian adults; the overwhelming majority of partially vaccinated at the time of the survey had later been boostered properly [18]. There was no reason to make distinction between partially and fully vaccinated persons. Therefore, participants were considered vaccinated if they had received at least one dose of the SARS-CoV-2 vaccine before the survey.

### 2.3. Explanatory Variables

Participants’ sex and age were registered along with their highest level of education. The applied age groups were 18–40, 41–70, and 71–105 years, to make distinction between the young, middle-aged and older adults. Primary, secondary and tertiary levels of education were distinguished. Employed and not-employed categories were applied. Non-regular alcohol drinkers, those consuming at least four drinks a day not more frequently than once a month, and regular alcohol drinkers with a more intense alcohol drinking history were identified. Regular and non-regular smoking dichotomization was used irrespective of the smoking intensity. The body mass index (BMI) was computed for each participant using self-declared weight and height data, and analyzed as a continuous variable. The presence or absence of chronic disease was registered regardless of the disease type.

### 2.4. Statistical Analysis

Chi-square and unpaired t-tests were used to assess the association between the explanatory and outcome variables. Multivariable logistic regression models were applied to describe the influence of sociodemographic parameters (age group, sex, and education), lifestyle (BMI, regular smoking, and regular alcohol drinking) and the presence of chronic disease on the outcome variables (test-confirmed SARS-CoV-2 infection, testing, contact tracing among infected individuals, and vaccination). The results were reported as odds ratios (OR) with corresponding 95% confidence intervals (95% CI).

The ISSP dataset was analyzed using SPSS version 26 (IBM SPSS Statistics for Windows, Version 26.0 Armonk, NY, USA.: IBM Corp.).

Ethical approval for the secondary analysis of anonymized data from ISSP was not required by Hungarian law.

## 3. Results

### 3.1. Descriptive Statistics of the Investigated Sample

Females, middle-aged adults (41–70 years old), employed persons, and secondary-educated adults predominated the investigated sample. This sociodemographic composition resembled Hungarian national reference values; however, the deviations were statistically significant apart from the employment ratio (Table 1).

The observed frequencies of regular smoking and alcohol drinking were 27.9% and 12.8%, respectively. The prevalence of at least one chronic disease was 27.2%, and 41.6% of the participants had a normal BMI. The European Health Interview Survey Hungarian Implementation dataset was used to generate reference numbers, and the investigated sample was representative with respect to the prevalence of regular smoking. Regular alcohol consumption and chronic disease were somewhat less prevalent among ISSP survey participants, and a normal BMI was somewhat more prevalent (Table 1).

The prevalence of test-confirmed SARS-CoV-2 infection was 12.1% in this sample. A total of 33.7% of participants were tested, and 67.9% of the infected participants were contact traced. The observed vaccination coverage was 52.4%. The reported ratios of test-confirmed infection, testing and vaccination were higher among ISSP survey responders than the communicated national reference values (Table 1).

### 3.2. Associations by Univariate Analyses

The univariate testing showed that there was no inequality in relation to sex and alcohol intake. Each outcome varied by age group. Persons at least 70 years old were less infected and less tested, but more vaccination occurred. In the survey, the primary-educated proved to be less infected, and less tested. While the tertiary educated were more tested and vaccinated. Employed adults were more infected, more tested, but less vaccinated. Vaccination was less intense among smokers. Persons with chronic disease were less tested, more traced, and more vaccinated. The BMI was higher among vaccinated than among not vaccinated (Table 2). (Results of the post-hoc analysis are summarized in the Appendix A).

### 3.3. Associations by Multivariate Analyses

According to the multivariable model, the occurrence of test-confirmed SARS-CoV-2 infection was not influenced by the investigated explanatory variables.

Testing was more frequent among middle-aged adults (aOR = 1.53, 95% CI 1.10–2.13) and employed adults (aOR = 2.06, 95% CI 1.42–3.00) and was more frequent among adults with a higher education than those with a primary education (aOR_secondary_ = 1.93, 95% CI 1.20–3.13; aOR_tertiary_ = 3.19, 95% CI 1.81–5.63).

Contact tracing was more frequently implemented among middle-aged (aOR_41–70y_ = 3.33, 95% CI 1.17–9.45) and employed (aOR = 4.58, 95% CI 1.38–15.22) adults and those with chronic diseases (aOR = 5.92, 95% CI 1.56–22.47).

Positive correlation was observed between age groups and vaccination frequency (aOR_41–70y_ = 2.94, 95% CI 2.09–4.15; aOR_71+y_ = 14.52, 95% CI 7.33–28.77). Higher than primary education (aOR_secondary_ = 1.69, 95% CI 1.08–2.63; aOR_tertiary_ = 4.36, 95% CI 2.46–7.73) and the presence of a chronic disease (aOR = 2.58, 95% CI 1.75–3.80) positively impacted vaccination. Regular smoking was inversely correlated with vaccination (aOR = 0.60; 95% CI 0.44–0.83) (Table 3).

## 4. Discussion

### 4.1. Main Findings

The SARS-CoV-2 infection rate in our study was 12%, slightly higher than the ECDC-reported indicators for Hungary (cumulative rate of infection: 8.2%), showing that our sample was not fully representative of Hungary.

The distribution of infections across the studied strata was not uneven in our investigation. The published stronger vulnerability of women [22,23,24,25], overweight persons [26,27,28], smokers [29,30], regular alcohol drinkers [29,30], employed [31,32,33] persons with low education [34,35], and patients with chronic diseases [31,36,37,38] was not observed in our investigation. The lack of these associations can be at least partly attributed to the low statistical power in our analysis.

The testing coverage of 33.7% observed in our survey was lower than in other high-income countries [39,40,41] although these reference data were reported considerably earlier than the ISSP survey. However, the Hungarian testing coverage was higher than that in Brazil [42] and India [43]. Unfortunately, Hungary ceased reporting testing frequency on 1 January 2021. Nevertheless, by the time of the ISSP survey the number of people tested was higher than the number of citizens in Europe. Therefore, the Hungarian testing frequency is considered to be below the European average.

Middle-aged [41,44,45], employed [41,44,45,46], and highly educated individuals [45,47] most frequently received testing, consistent with published international experiences [48]. It is the consequence of the adaption of the high-risk target groups’ definition for screening group by the WHO recommendations [49]. The influencing impact of being overweight, smoking, and alcohol consumption on SARS-CoV-2 testing [47] was not observed in our sample. Furthermore, we found no association between chronic diseases [50,51] and SARS-CoV-2 testing.

Altogether, the low testing rates were accompanied by a relatively high infection rate in Hungary. This gap could contribute to the higher than expected mortality rate. Moreover, this problem could be more pronounced among less educated, unemployed, and non-middle-aged adults.

The coverage of contact tracing among infected individuals was 74.8% in our survey, far lower than that reported in Spain, Hong Kong, and India [52,53,54] but similar to that reported in Catalonia [55]. The low incidence of contact tracing could be attributed to the failure of laboratories to communicate positive test results to public health authorities, the lack of contact information provided in the notifications, the refusal of patients to participate, and the lack of contact tracing for patients admitted to the hospital. Although the number of contact tracing staff increased enormously in Hungary, involving police and municipalities and governmental department employees, more contact tracers should have been employed. A contact tracing team 1.34 times larger than the one used would have been beneficial as it could ensure reaching all infected instead of just 74.8% of them.

The inequalities in contact tracing implementation in Hungary were comparable to those in other countries; contact tracing was more prevalent among employed adults [56,57,58,59], middle-aged individuals [56,57], older adults [48,60], highly educated individuals [61], and patients with chronic disease [60]. No association between smoking, high BMI, and regular alcohol consumption with contact tracing was observed. As contact tracing is crucial in preventing infection spread, incomplete contact tracing may have contributed to excess mortality. This impact could be more profound among non-middle-aged, unemployed, and primary-educated adults, and those without chronic disease.

The vaccination rate in our study was 52.4% (95% CI: 49.3–55.5), slightly higher than the ECDC-reported indicators for Hungary (50.9%) and significantly higher than that reported in the EU (44%). The high vaccination rate was the main positive characteristic of Hungarian epidemic control at the time of the ISSP survey. The mass vaccination program started in January 2021; thus, vaccination likely reduced COVID-19-related mortality in Hungary before the ISSP survey was conducted. Therefore, the effect of infrequent testing and poor contact tracing practices is likely more serious than suggested by the 11,837 rate-corrected excess cases.

Middle-aged, older, employed adults, those with higher education [61,62,63], and those with chronic diseases [60,64,65] were more likely to be vaccinated in our study sample, similar to the findings of other studies. It has been demonstrated that being obese is a significant factor in the likelihood of adverse immune responses to SARS-CoV-2 vaccination [66,67]; however, this observation is contrary to our findings. Regular smokers were considerably less likely to be vaccinated against SARS-CoV-2, consistent with other published studies [68,69].

### 4.2. Strengths and Limitations

The data collection was organized by an ISSP team with decades of experience, and the data were inserted into the omnibus survey of an international project with high quality standards.

Although each outcome was assessed by self-reporting, the participants were asked about their recent, 12 to 14 months’ experiences regarding a very important issue. Therefore, significant recall bias was unlikely.

Males, middle-aged individuals, secondary-educated individuals, regular drinkers, and individuals with chronic diseases were underrepresented in the sample. Although the representativeness was not perfect, deviations were less than 10% apart from the proportion of secondary-educated (66.8% in the sample, 55.0% in the population). The relatively small sample size prevented a more detailed analysis evaluating the role of potential influencing factors (more detailed socioeconomic and clinical indicators) and resulted in results with borderline significance. A more detailed investigation with a larger sample size is required to improve the interpretation of our study.

All persons with infections were not identified in the survey due to the low testing activity. There were subjects with subclinical infection without perception of the infection. They were not tested because the test-based, organized screening program was restricted to selected high-risk populations (defined by occupation) in Hungary. Additionally, it was directly quantified by the survey that there were persons with recognized symptoms but there was no test confirmation. Consequently, persons with infection confirmed by laboratory findings, who were considered as infected in our analyses, were composed of a subgroup of all infected persons.

### 4.3. Implications

The populations of unemployed and primary-educated people are 5.3 and 3 million, respectively, in Hungary. Converting the adjusted odds ratio into the adjusted risk ratio [70], the estimated number of people without contact tracing among unemployed individuals was 240,715, the number of unvaccinated primary-educated individuals was 1,076,541, and the numbers of primary-educated and unemployed individuals who did not receive testing were 647,828 and 1,332,869, respectively. Clearly, these missing interventions should be prevented.

Smokers’ risk-taking behavior [68,71] and smokers believe that they are protected against the severe manifestation of COVID-19 infection [72] can be reflected in their lower vaccination rate. The organization of vaccination should seriously consider the special needs of this vulnerable group because it is well-known that the consequences of COVID-19 infection are more serious among smokers than among non-smokers [73].

Patients with chronic disease provided with continuous care were not traced and vaccinated rigorously according to the guidelines, which is unacceptable from a quality-management viewpoint. While the less-than-complete vaccination coverage can be explained by the restricted availability of vaccines at the time of the survey, the lack of contact tracing is an obvious failure of epidemiological control.

Our findings suggest that more rigorous organization of epidemiological measures is needed. To establish better organization, more intensive monitoring is also needed. It is insufficient to monitor testing, infection rate, contact tracing, and vaccination rate by aggregated indicators. Detailed monitoring is required to identify high-risk groups receiving substandard care. The epidemiological measure of authorities, healthcare providers, and common people were—in many occasions desperate—reactions to an exceptional challenge, which were based in many cases on expert opinion during the COVID-19 pandemic. Honestly, a lot of decisions lacked the usual evidence base [74]. As the expert opinions relied on weak evidence and on uncertain quality data, it was inevitable that the lack of knowledge evolved the room for prevailing other interests from the political arena, economy, and mass media. Expert opinions incorporated both the insufficient scientific knowledge and the non-scientific influences. However, the nature of the epidemic urged interventions which deeply intervene into the normal operations of the society. During the emergent situation where high-impact decisions and their consistent implementation were needed in spite of the fact that the evidence was insufficient, it was obvious that some decisions were not proper. Unfortunately, the inappropriateness of a decision could be perceived only after the manifestation of the consequences not intended. It was unavoidable to undertake the learning-by-doing approach [75]. To improve our ability to realize the need for modification of a decision, more timely and detailed monitoring is required, which makes it possible to improve the effectiveness of the epidemiological control measures at any level.

## 5. Conclusions

The survey was conducted during the third wave of the COVID-19 pandemic, when the mass vaccination started, and the pandemic control mainly relied on epidemiologic measures. (1) It was the first Hungarian study on person-reported experiences about epidemiological control measures. (2) The socioeconomic or lifestyle-related inequalities in test-confirmed SARS-CoV-2 infections were not confirmed in Hungary. However, the survey indicated that testing, contact tracing, and vaccination were seriously influenced by socioeconomic position and less so by chronic disease prevalence and very minimally by lifestyle. (3) Considering that the socioeconomic inequalities in COVID-19-related deaths have been demonstrated convincingly in Hungary and that epidemic measures are obviously effective, the etiological role of socioeconomic inequalities in epidemic measure implementation likely generated socioeconomic inequality in COVID-19-related death rates. Therefore, monitoring sensitively to the target group characteristics when implementing epidemic measures appears necessary in managing epidemiological measures, and socially adapted implementation can mitigate the health effects of the COVID-19 epidemic.

## Figures and Tables

**Table 1 healthcare-11-01220-t001:** Sociodemographic composition, lifestyle factors and chronic disease occurrence of the sample investigated in the ISSP 2021 survey with reference data from the Hungarian population.

Influencing Factor		ISSP Survey (95% Confidence Interval)	Hungarian Reference (95% Confidence Interval)
Age *	18–40 years	26.8% (24.6–29.5)	23.2%
	40–70 years	60.1% (57.1–63.1)	55.0%
	71–105 years	13.1% (11.1–15.1)	21.8%
Sex *	Male	40.9% (37.9–43.9)	47.5%
	Female	59.1% (56.1- 62.1)	52.5%
Education *	Primary	16.4% (14.1–18.7)	23.2%
	Secondary	68.8% (66.0–71.7)	55.0%
	Tertiary	14.8% (12.6–17.0)	21.8%
Employment *	Employed	65.2% (62.1–68.1)	64.0%
	Not-employed	34.6% (31.7–37.6)	36.0%
Smoking **	Regular smoking	27.9% (25.1–30.6)	26.3% (25.1–27.4)
	Not regular smoking	72% (69.3–74.8)	81.8% (80.6–83.0)
Drinking alcohol **	Regular drinking	12.8% (10.7–14.8)	18.2% (17.0–19.4)
	Non-regular drinking	87.1% (85.1–89.2)	81.8% (80.6–83.0)
BMI **	Thin	4.1% (2.8–5.3)	2.5% (2.1–2.9)
	Normal	41.6 % (38.5–44.7)	36.8% (35.5–38.0)
	Overweight	37.2 (34.2–40.3)	34.6% (33.3–35.8)
	Obese	17.1 % (14.7–19.5)	24.8% (23.7–25.9)
Chronic Diseases **	Present	27.2% (24.5–30.0)	34.8% (31.9–37.8)
	Absent	72.8% (70.0–75.5)	65.2% (62.2–68.1)
Infection ***	(test-confirmed)	12.1% (10.1–14.2)	8.2%
Testing	(under any circumstances)	33.7% (30.8–36.6)	#
Tracing	(among infected)	74.8% (67.0–82.6)	##
Vaccination ***	(by at least one dose)	52.4% (49.3 -55.5)	50.9%

* Reference: Hungarian Mid-term Census, 2016 [19]. ** Reference: Hungarian implementation of the European Health Interview Survey, 2019 [20]. *** Reference: European Centre for Disease Prevention and Control, 2021 [21]. # Hungary discontinued reporting on testing frequency at the end of 2020 to the ECDC. There is no national reference value for the study period. ## Contact tracing was not included in the ECDC monitoring system.

**Table 2 healthcare-11-01220-t002:** Prevalence of SARS-CoV-2 infection and the implementation of epidemiological measures by sociodemographic status, lifestyle and chronic disease occurrence in the sample investigated by the ISSP 2021 survey.

		Distribution	Infected ^#^*	*p* ***	Testing *	*p* ***	Traced *	*p* ***	Vaccinated *	*p* ***
Age	18–40 year	270 (26.8%)	32/262 (12.2%)	0.006	85/270 (31.5%)	<0.001	18/32 (56.3%)	0.031	77/270 (28.5%)	<0.001
	41–70 year	606 (60.1%)	82/590 (13.9%)		238/606 (39.3%)		67/82 (81.7%)		338/606 (55.8%)	
	71- year	132 (13.1%)	5/130 (3.8%)		17/132 (12.9%)		4/5 (80%)		113/132 (85.6%)	
Sex	Female	596 (59.1%)	67/582 (11.5%)	0.483	205/596 (34.4%)	0.591	51/67 (76.1%)	0.597	324/596 (54.4%)	0.130
	Male	412 (40.9%)	52/400 (13%)		135/412 (32.8%)		38/52 (73.1%)		204/412 (49.5%)	
Education	Primary	165 (16.4%)	9/161 (5.6%)	0.021	26/165 (15.8%)	<0.001	6/9 (66.7%)	0.324	87/165 (52.7%)	<0.001
	Secondary	694 (68.8%)	90/675 (13.3%)		241/694 (34.7%)		68/90 (75.6%)		337/694 (48.6%)	
	Tertiary	149 (14.8%)	20/146 (13.7%)		73/149 (49%)		15/20 (75%)		104/149 (69.8%)	
Employment	Employed	657 (65.2%)	90/638 (14.1%)	0.007	269/657 (40.9%)	<0.001	69/90 (76.7%)	0.279	312/657 (47.5%)	<0.001
	Not-employed	349 (34.6%)	28/342 (8.2%)		70/349 (20.1%)		19/28 (67.9%)		216/349 (61.9%)	
	Missing	2 (0.2%)	1/2 (50%)		1/2 (50%)		1/1 (100%)		0/2 (0%)	
Smoking	Regular	281 (27.9%)	34/278 (12.2%)	0.932	96/281 (34.2%)	0.764	27/34 (79.4%)	0.943	112/281 (39.9%)	<0.001
	Not smoker	726 (72%)	85/703 (12.1%)		244/726 (33.6%)		62/85 (72.9%)		416/726 (57.3%)	
	Missing	1 (0.1%)	0/1 (0%)		0/1 (0%)		---		0/1 (0%)	
Alcohol drinking	Regular	129 (12.8%)	17/126 (13.5%)	0.770	39/129 (30.2%)	0.394	13/17 (76.5%)	0.760	61/129 (47.3%)	0.149
	Non-regular	877 (87%)	102/854 (11.9%)		301/877 (34.3%)		76/102 (74.5%)		467/877 (53.2%)	
	Missing	2 (0.2%)	0/2 (0%)		0/2 (0%)		---		0/2 (0%)	
Chronic disease	Present	274 (27.2%)	30/265 (11.3%)	0.835	74/274 (27%)	0.017	28/30 (93.3%)	0.036	203/274 (74.1%)	<0.001
	Absent	733 (72.7%)	89/716 (12.4%)		266/733 (36.3%)		61/89 (68.5%)		325/733 (44.3%)	
	Missing	1 (0.1%)	0/1 (0%)		0/1 (0%)		---		0/1 (0%)	
BMI **	Mean ± SD	26.1 ± 4.5	26.1 ± 4.6/26.0 ± 4.5	0.832	25.8 ± 4.5/26.2 ± 4.5	0.229	26.64 ± 4.85/24.73 ± 3.73	0.431	26.4 ± 4.3/25.7 ± 4.6	0.008
Total		1008 (100%)	119/982 (12.1%)		340/1008 (33.7%)		89/119 (74.8%)		528/1008 (52.4%)	

* number of cases with positive outcome/total number of cases (percentage of positive outcome). ** mean ± SD among outcome positives/mean ± SD among outcome negatives. *** chi-square test, and *t*-test for BMI. ^#^ infections confirmed by laboratory findings.

**Table 3 healthcare-11-01220-t003:** Associations with sociodemographic and lifestyle factors and the presence of chronic disease with test-confirmed SARS-CoV-2 infection, testing for SARS-CoV-2 infection, contact tracing, and SARS-CoV-2 vaccination according to multivariable logistic regression analyses (adjusted odds ratios with 95% confidence intervals).

Influencing Factors		Infection	Testing	Tracing *	Vaccination
Sex	Male/Female	1.12 (0.73–1.73)	1.04 (0.77–1.42)	0.50 (0.20–1.26)	0.85 (0.62–1.16)
Age (years)	41–70/18–40	1.15 (0.71–1.84)	1.53 (1.10–2.13)	3.33 (1.17–9.45)	2.94 (2.09–4.15)
	71+/18–40	0.37 (0.12–1.10)	0.70 (0.36–1.37)	4.60 (0.45–46.89)	14.52 (7.33–28.77)
Education	Secondary/Primary	2.00 (0.94–4.25)	1.93 (1.20–3.13)	3.95 (0.96–16.24)	1.69 (1.08–2.63)
	Tertiary/Primary	2.11 (0.88–5.06)	3.19 (1.81–5.63)	4.73 (0.85–26.37)	4.36 (2.46–7.73)
Employment	Employed/Not-employed	1.43 (0.84–2.43)	2.06 (1.42–3.00)	4.58 (1.38–15.22)	1.25 (0.87–1.79)
Drinking Alcohol	Regular drinking/Non-regular drinking	1.17 (0.65–2.11)	0.83 (0.53–1.28)	1.08 (0.33–3.46)	0.90 (0.58–1.40)
Smoking	Regular smoking/Not regular smoking	0.87 (0.55–1.36)	0.99 (0.72–1.35)	0.96 (0.37–2.50)	0.60 (0.44–0.83)
BMI	kg/m^2^	1.00 (0.96–1.05)	0.98 (0.95–1.02)	0.99 (0.90–1.09)	0.99 (0.96–1.03)
Chronic Disease	Present/Absent	1.43 (0.84–2.41)	1.17 (0.80–1.71)	5.92 (1.56–22.47)	2.58 (1.75–3.80)

* among test positive infected.

## Data Availability

The database of EHIS 2019 is available from the Hungarian Central Statistical Office through a registration process (https://www.ksh.hu/data_access_safe_centre_access, accessed on 8 March 2023). Data downloadable version of the International Social Survey Program 2021 is under preparation (http://www.issp.org/data-download/by-year/, accessed on 20 January 2023). The database of ECDC 2021 is available from https://www.ecdc.europa.eu/en/covid-19/data, accessed on 20 January 2023.

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
