# Peer review of "Effectiveness of and Inequalities in COVID-19 Epidemic Control Strategies in Hungary: A Nationwide Cross-Sectional Study"

_healthcare, 2023, doi:10.3390/healthcare11091220_

Round 1
Reviewer 1 Report
Overall, this article is well written and presents the research findings clearly and concisely. The conclusions drawn from this study provide important insights into the impact of sociodemographic, lifestyle, and clinical factors on COVID-19-related epidemic control measures. However, there are a few areas where the language could be improved or clarified:
- The sentence "Data for the secondary analysis were obtained from the 2021 International Social Survey Program (ISSP) conducted from March 15 to May 30, 2021, during the third wave of COVID-19" is a bit confusing. It is not clear what is meant by "secondary analysis", and it may be useful to explain it more explicitly. Additionally, clarifying whether ISSP surveys are specifically designed to collect data on COVID-19 or whether COVID-19-related questions added as additional modules to existing surveys may be useful.
- The phrase "randomly selected from the country’s whole population" . How you randomly select respondents, what sampling techniques are used to ensure population representation
- You say vaccination rates in smokers are lower. Further information to explain this phenomenon may be important for health policy actors in intervening in vaccination coverage in these groups
Author Response
Dear Reviewer,
Thank you very much for the careful review of our manuscript. Please find enclosed the revised version of the manuscript “Effectiveness of and inequalities in COVID-19 epidemic control strategies in Hungary: A nationwide cross-sectional study” by János Sándor, et al.
Each comment and suggestion has been considered. The corresponding changes and refinements made in the revised paper are summarized in our response after considering each of your suggestion. Answers along with the modifications we made are summarized below (comments/questions of Yours are in capitals).
Sincerely yours, Janos Sandor (on behalf of the authors)
Answers/reflections to the comments of Reviewer-1:
1.
OVERALL, THIS ARTICLE IS WELL WRITTEN AND PRESENTS THE RESEARCH FINDINGS CLEARLY AND CONCISELY. THE CONCLUSIONS DRAWN FROM THIS STUDY PROVIDE IMPORTANT INSIGHTS INTO THE IMPACT OF SOCIODEMOGRAPHIC, LIFESTYLE, AND CLINICAL FACTORS ON COVID-19-RELATED EPIDEMIC CONTROL MEASURES.
Thank you very much for this evaluation!
HOWEVER, THERE ARE A FEW AREAS WHERE THE LANGUAGE COULD BE IMPROVED OR CLARIFIED:
2.
THE SENTENCE "DATA FOR THE SECONDARY ANALYSIS WERE OBTAINED FROM THE 2021 INTERNATIONAL SOCIAL SURVEY PROGRAM (ISSP) CONDUCTED FROM MARCH 15 TO MAY 30, 2021, DURING THE THIRD WAVE OF COVID-19" IS A BIT CONFUSING. IT IS NOT CLEAR WHAT IS MEANT BY "SECONDARY ANALYSIS", AND IT MAY BE USEFUL TO EXPLAIN IT MORE EXPLICITLY. ADDITIONALLY, CLARIFYING WHETHER ISSP SURVEYS ARE SPECIFICALLY DESIGNED TO COLLECT DATA ON COVID-19 OR WHETHER COVID-19-RELATED QUESTIONS ADDED AS ADDITIONAL MODULES TO EXISTING SURVEYS MAY BE USEFUL.
The sentence in 2.1 Setting has been corrected: it was declared explicitly that COVID-19 epidemic related questions had been added to the ISSP survey questions. Further, the “secondary analysis” term was deleted.
Original sentence:
Data for the secondary analysis were obtained from the 2021 International Social Sur-vey Program (ISSP) conducted from March 15 to May 30, 2021, during the third wave of COVID-19.
Corrected sentence (Line 83-85):
Data were obtained from the 2021 International Social Survey Program (ISSP) con-ducted from March 15 to May 30, 2021, during the third wave of COVID-19. The survey was completed with COVID-19 epidemic related questions.
3.
THE PHRASE "RANDOMLY SELECTED FROM THE COUNTRY’S WHOLE POPULATION" . HOW YOU RANDOMLY SELECT RESPONDENTS, WHAT SAMPLING TECHNIQUES ARE USED TO ENSURE POPULATION REPRESENTATION.
The survey generated a sample, which was representative of the country’s whole adult (above-18-population) population. Random selection was carried out using the national register of the resident Hungarian population as sampling frame.
Original sentence:
A representative sample of 1008 Hungarian adults over 18 years of age was randomly selected from the country’s whole population and inter-viewed.
Corrected sentence (Line 87-89):
A representative sample of 1008 Hungarian adults over 18 years of age was randomly selected from the country’s whole population (using the national registry of the Hungarian population as sampling frame) and interviewed by trained interviewers
4.
YOU SAY VACCINATION RATES IN SMOKERS ARE LOWER. FURTHER INFORMATION TO EXPLAIN THIS PHENOMENON MAY BE IMPORTANT FOR HEALTH POLICY ACTORS IN INTERVENING IN VACCINATION COVERAGE IN THESE GROUPS.
Thank you very much for this comment. We agree with you that the lower vaccination rate among smokers should be considered by the public health authorities responsible for vaccination organization, because smokers are at higher risk of the serious outcomes of the COVID-19 infection than nonsmokers. The corresponding paragraph has been completed with more argumentation and the associated references.
Original sentence:
Smokers’ risk-taking behavior [70] is reflected in their lower vaccination rate. The organization of vaccination should consider the special needs of this vulnerable group.
Corrected sentence (Line 289-293):
Smokers’ risk-taking behavior [73,74] and smokers’ believe that they are protected against the severe manifestation of COVID-19 infection [72] can be reflected in their lower vaccination rate. The organization of vaccination should seriously consider the special needs of this vulnerable group, because it is well known that the consequences of COVID-19 infection are more serious among smokers than among nonsmokers [75].

Reviewer 2 Report
The study goal is to describe patient experiences with COVID-19 related testing, contact tracing, and vaccination. The article also evaluates the impact of sociodemographic, lifestyle, and clinical factors on patient-reported epidemiological measures. Last, the goal is to evaluate the real world effectiveness of the epidemiological control.
This is an interesting study highlighting the mortality excess rate in Hungary during the COVID-19 pandemic. The study was performed during the covid 3rd wave. I believe this should be clearly indicated at the beginning rather than at the end of the article.
One of the main drawbacks I see in this article is that it lacks a literature review to start with. Indeed, have there been other covid-19 related studies in Hungary, in Europe ? What are the results ? We should be able to contextualize this study better.
Line 40 : I guess there is a mistake in year 2021
Line 43 : the acronym ECDC should be explained
Line 86 : Participants were considered vaccinated if they had received at least one dose of the SARS-CoV-2 vaccine before 86 the survey. I am not sure of the regulation in Hungary but in many other country, one single dose was not considered as "vaccinated". If the dataset includes the numbers of received doses by participants, the analysis could distinguish between the number of doses.
Line 90 : how are explained the chosen age groups ?
Table 2 : change the editing (increase column size for the P-values)
Line 203 : "contact" tracing
Author Response
Dear Reviewer,
Thank you very much for the careful review of our manuscript. Please find enclosed the revised version of the manuscript “Effectiveness of and inequalities in COVID-19 epidemic control strategies in Hungary: A nationwide cross-sectional study” by János Sándor, et al.
Each comment and suggestion has been considered. The corresponding changes and refinements made in the revised paper are summarized in our response after considering each of your suggestion. Answers along with the modifications we made are summarized below (comments/questions of Yours are in capitals).
Sincerely yours, Janos Sandor (on behalf of the authors)
Answers/reflections to the comments of Reviewer-2:
1.
THE STUDY GOAL IS TO DESCRIBE PATIENT EXPERIENCES WITH COVID-19 RELATED TESTING, CONTACT TRACING, AND VACCINATION. THE ARTICLE ALSO EVALUATES THE IMPACT OF SOCIODEMOGRAPHIC, LIFESTYLE, AND CLINICAL FACTORS ON PATIENT-REPORTED EPIDEMIOLOGICAL MEASURES. LAST, THE GOAL IS TO EVALUATE THE REAL WORLD EFFECTIVENESS OF THE EPIDEMIOLOGICAL CONTROL.
Thank you for this summary.
2.
THIS IS AN INTERESTING STUDY HIGHLIGHTING THE MORTALITY EXCESS RATE IN HUNGARY DURING THE COVID-19 PANDEMIC. THE STUDY WAS PERFORMED DURING THE COVID 3RD WAVE. I BELIEVE THIS SHOULD BE CLEARLY INDICATED AT THE BEGINNING RATHER THAN AT THE END OF THE ARTICLE.
Actually, our investigation was initiated by the extreme excess mortality during the COVID-19 epidemic in Hungary. We wanted to understand the role of the non-perfect epidemiologic control measures in this excess mortality.
The fact that the excess mortality in Hungary during the COVID-19 epidemic was high in Hungary compared to other European countries is mentioned in the introduction. (Line 69-71)
It is mentioned in the 2.1 Setting that the data collection was carried out during the third wave of the COVID-19 epidemic. (Line 85)
In order to make more explicit the initiation of our investigation, the objectives was corrected.
Original sentence:
Because compliance with epidemiological measures has not been evaluated yet in Hungary, our study aimed (1) to describe patient experiences with COVID-19 related testing, contact tracing, and vaccination, (2) to evaluate the impact of sociodemographic, lifestyle, and clinical factors on patient-reported epidemiological measures, and (3) to evaluate the real-world effectiveness of the epidemiological control.
Corrected sentence (Line 76-80):
Because compliance with epidemiological measures has not been evaluated yet in Hungary, our study aimed (1) to describe patient experiences with COVID-19 related testing, contact tracing, and vaccination, (2) to evaluate the impact of sociodemographic, lifestyle, and clinical factors on patient-reported epidemiological measures, and (3) the contribution of non-perfect epidemiological control measures to the mortality excess.
3.
ONE OF THE MAIN DRAWBACKS I SEE IN THIS ARTICLE IS THAT IT LACKS A LITERATURE REVIEW TO START WITH. INDEED, HAVE THERE BEEN OTHER COVID-19 RELATED STUDIES IN HUNGARY, IN EUROPE? WHAT ARE THE RESULTS? WE SHOULD BE ABLE TO CONTEXTUALIZE THIS STUDY BETTER.
To extend the description of background of the investigation, we added more references to the introduction.
- Nagy É, Golopencza P, Barcs I, Ludwig E. Comparison of COVID-19 Severity and Mortality Rates in the First Four Epidemic Waves in Hungary in a Single-Center Study with Special Regard to Critically Ill Patients in an Intensive Care Unit. Trop Med Infect Dis. 2023 Mar 1;8(3):153. doi: 10.3390/tropicalmed8030153
- Balint L, Osvath P, Kapitany B, Rihmer Z, Nemeth A, Dome P. Suicide in Hungary during the first year of the COVID-19 pandemic: Subgroup investigations. J Affect Disord. 2023 Mar 15;325:453-458. doi: 10.1016/j.jad.2023.01.046
- Lantos T, Nyári TA. The impact of the COVID-19 pandemic on suicide rates in Hungary: an interrupted time-series analysis. BMC Psychiatry. 2022 Dec 9;22(1):775. doi: 10.1186/s12888-022-04322-2
- Sobczak M, Pawliczak R. COVID-19 mortality rate determinants in selected Eastern European countries. BMC Public Health. 2022 Nov 16;22(1):2088. doi: 10.1186/s12889-022-14567-x
4.
LINE 40 : I GUESS THERE IS A MISTAKE IN YEAR 2021
Thanks for this notice! Corrected as suggested.
Original sentence:
As of May 3, 12021, …
Corrected sentence (Line 43-44):
As of May 3, 2021, …
5.
LINE 43 : THE ACRONYM ECDC SHOULD BE EXPLAINED
Thanks for this notice! Corrected as suggested.
Original sentence:
The ECDC estimates that …
Corrected sentence (Line 47-48):
The European Center for Disease Prevention and Control (ECDC) estimates that …
6.
LINE 86: PARTICIPANTS WERE CONSIDERED VACCINATED IF THEY HAD RECEIVED AT LEAST ONE DOSE OF THE SARS-COV-2 VACCINE BEFORE THE SURVEY. I AM NOT SURE OF THE REGULATION IN HUNGARY BUT IN MANY OTHER COUNTRY, ONE SINGLE DOSE WAS NOT CONSIDERED AS "VACCINATED". IF THE DATASET INCLUDES THE NUMBERS OF RECEIVED DOSES BY PARTICIPANTS, THE ANALYSIS COULD DISTINGUISH BETWEEN THE NUMBER OF DOSES.
There were 6 types of vaccine used in Hungary [Janssen (Johnson and Johnson, New Brunswick, NJ, USA), Moderna (National Institute of Allergy and Infectious Diseases and Biomedical Advanced Research and Development Authority, Cambridge, MA, USA), AstraZeneca (Oxford University, Oxford, UK), Pfizer (BioNTech, Mainz, Germany), Sinopharm (Wuhan Institute of Biological Products, Wuhan, China), and Sputnik (Gamaleya National Research Centre of Epidemiology and Microbiology, Moscow, Russia)]. The Hungarian protocol for COVID-19 vaccination was in concordance with the international recommendation at the time of the survey. The first dose of vaccine had to be followed by the booster (apart from the Janssen vaccine).
The mass vaccination in the period of the data collection was in progress. As it was published, the persons who partially vaccinated at 1 April 2021 was 1,033,264 (among the 7,437,966 target population of the Hungarian adults;14%) Later, due to the continuation of the mass vaccination, this partly vaccinated population reduced to 189,568 (3%) by 21 June 2021. Altogether, the partial vaccination was rare. We added the reference for these data. [Pálinkás A, Sándor J. Effectiveness of COVID-19 Vaccination in Preventing All-Cause Mortality among Adults during the Third Wave of the Epidemic in Hungary: Nationwide Retrospective Cohort Study. Vaccines (Basel). 2022 Jun 24;10(7):1009. doi: 10.3390/vaccines10071009]
Therefore, the vaccination status was not assessed in details in the survey.
Original sentence:
Participants were considered vaccinated if they had received at least one dose of the SARS-CoV-2 vaccine before the survey.
Corrected sentence (Line 98-104):
The mass vaccination was in progress in the time of the data collection, and the partial vaccination was rare among Hungarian adults, the overwhelming majority of partially vaccinated in the time of the survey had been later boostered properly [18]. There was no reason to make distinction between partially and fully vaccinated persons. Therefore, participants were considered vaccinated if they had received at least one dose of the SARS-CoV-2 vaccine before the survey.
7.
LINE 90: HOW ARE EXPLAINED THE CHOSEN AGE GROUPS?
Our intention was to make distinction between young, middle aged and older adults. We choose arbitrary the 40 years and 70 years as thresholds for the categorization.
Original sentence:
The applied age groups were 18-40, 41-70, and 71-105 years.
Corrected sentence (Line 107-108):
The applied age groups were 18-40, 41-70, and 71-105 years, to make distinction be-tween the young, middle aged and older adults.
8.
TABLE 2: CHANGE THE EDITING (INCREASE COLUMN SIZE FOR THE P-VALUES)
Thanks for this notice! Corrected as suggested.
9.
LINE 203: "CONTACT" TRACING
Thanks for this notice! Corrected as suggested.
Original sentence:
The coverage of contract tracing among infected individuals …
Corrected sentence (Line 223):
The coverage of contact tracing among infected individuals …

Reviewer 3 Report
Overall
The article deals with an interesting topic and gives some insights into the Hungarian experience. Overall, it is written in a comprehensive way, but there are some gaps and incomprehensible conclusions. Especially the methods and results should be described more thoroughly. I would suggest, that the article can only be published after major revision.
Abstract
l. 16/ l. 77: you write „interviewed, but in other text passages it sounds like the participants participated in an online survey? It is unclear how the ISSP survey was conducted. Please then also go more in detail about the recruiting of the participants, limitations etc.
l. 26/161 Please check if dose-response association is the right term.
Introduction
l. 62 What do you mean with settlement level deprivation in this context?
l. 67 You did not really describe the experiences of the patients (e.g. if it was easy to get a test). Please rephrase
Methods
l. 79-98: Why did you choose to check those specific variables? Were they the only ones available or were there others reasons? Please be more specific.
l. 90: Why did you choose these age groups? They seem quite random.
Results
Please include subheadings to structure the results part.
l. 137-143 please report the p values in the text.
l. 138 please define elderies, which age group is this? The same goes for more or less educated particpiants.
l. 140 when you get tested less, you probably notice less infections. Please include this in the limitations.
l. 142 was associated how? Positive or negative?
Table 2. Why is the denominator of distribution and infected not the same?
How did you test when you had more than two expressions? E. g. age, to which groups does the p-value refer?
Discussion
Please give a short overview about the main findings at the beginning of the discussion.
l. 194 How can this be consistent with the recommendations? Is it recommend to e.g. test highly educated individuals more often? Please rephrase.
l. 211 Why 1.5 times larger? Where do you get this number from?
l. 219 Please be careful with assumptions. You can only guess that the impact might be more profound, not that it is.
l. 231 How did you observe if the obese population had an adverse immune response from your data?
l. 241 What time period did they have to recall? Around 6 month since the beginning of the pandemic?
l. 259 Is the risk taking behavior the only possible explanation for a lower vaccination rate? What about the interaction with lower educational level etc.?
Conclusion
Please refer back to the study aims and if/how you completed them.
Author Response
Dear Reviewer,
Thank you very much for the careful review of our manuscript. Please find enclosed the revised version of the manuscript “Effectiveness of and inequalities in COVID-19 epidemic control strategies in Hungary: A nationwide cross-sectional study” by János Sándor, et al.
Each comment and suggestion has been considered. The corresponding changes and refinements made in the revised paper are summarized in our response after considering each of your suggestion. Answers along with the modifications we made are summarized below (comments/questions of Yours are in capitals).
Sincerely yours, Janos Sandor (on behalf of the authors)
Answers/reflections to the comments of Reviewer-3:
1.
OVERALL THE ARTICLE DEALS WITH AN INTERESTING TOPIC AND GIVES SOME INSIGHTS INTO THE HUNGARIAN EXPERIENCE. OVERALL, IT IS WRITTEN IN A COMPREHENSIVE WAY, BUT THERE ARE SOME GAPS AND INCOMPREHENSIBLE CONCLUSIONS. ESPECIALLY THE METHODS AND RESULTS SHOULD BE DESCRIBED MORE THOROUGHLY. I WOULD SUGGEST, THAT THE ARTICLE CAN ONLY BE PUBLISHED AFTER MAJOR REVISION.
Thank you very much for this evaluation!
2.
Abstract
- 16/ L. 77: YOU WRITE „INTERVIEWED, BUT IN OTHER TEXT PASSAGES IT SOUNDS LIKE THE PARTICIPANTS PARTICIPATED IN AN ONLINE SURVEY? IT IS UNCLEAR HOW THE ISSP SURVEY WAS CONDUCTED. PLEASE THEN ALSO GO MORE IN DETAIL ABOUT THE RECRUITING OF THE PARTICIPANTS, LIMITATIONS ETC.
The survey was not on-line. Trained interviewers reported the selected persons.
Original sentence:
A representative sample of 1008 Hungarian adults over 18 years of age was randomly selected from the country’s whole population and inter-viewed.
Corrected sentence (Line 87-89):
A representative sample of 1008 Hungarian adults over 18 years of age was randomly selected from the country’s whole population (using the national registry of the Hungarian population as sampling frame) and interviewed by trained interviewers
3.
- 26/161 PLEASE CHECK IF DOSE-RESPONSE ASSOCIATION IS THE RIGHT TERM.
The misleading term has been replaced both in the abstract and in the main text.
Original sentence:
A dose‒response association was observed between age groups and vaccination frequency
Corrected sentence (Line 28-29 and Line 183):
Positive correlation was observed between age groups and vaccination frequency
4.
Introduction
- 62 WHAT DO YOU MEAN WITH SETTLEMENT LEVEL DEPRIVATION IN THIS CONTEXT?
Hungarian settlements (towns and cities) were the subjects of the studies referred. The association between settlement level aggregated indicators were analyzed. The relationship of vaccination coverage and mortality rates with a composite indicator of the socioeconomic status of the population living in the settlements were correlated. It was an ecological study. To emphasize the design of that study, the sentence has been corrected accordingly.
Original sentence:
The socio-economic position influenced the epidemic in Hungary.
Corrected sentence (Line 72-73):
According to ecological studies, the socio-economic position influenced the epidemic in Hungary.
5.
- 67 YOU DID NOT REALLY DESCRIBE THE EXPERIENCES OF THE PATIENTS (E.G. IF IT WAS EASY TO GET A TEST). PLEASE REPHRASE
Maybe, we do not understand your criticism. During the survey, participants reported to the interviewers whether they had symptoms, underwent testing, were subject of contact tracing, or had been vaccinated. This part of the data collection was a patient reported experience survey. We hope that you can accept this argumentation. Text was not corrected.
6.
METHODS
- 79-98: WHY DID YOU CHOOSE TO CHECK THOSE SPECIFIC VARIABLES? WERE THEY THE ONLY ONES AVAILABLE OR WERE THERE OTHERS REASONS? PLEASE BE MORE SPECIFIC.
As it is written in the introduction, the social gradient in COVID-19 death rates were demonstrated in Hungary by ecological investigation. Our main question was whether the social gradient of epidemiological control measures’ effectiveness did contribute to that mortality gradient. Before to check it, we had to describe the person level experiences on epidemiological control measures, and then we had to determine the social gradient for those outcomes. Because it was known that the lifestyle and the clinical status have impact on the infection, testing, contact tracing, and vaccination, we carried out the investigation with controlling for confounding effects by inserting proxy measures for life-style and clinical status into the regression models. We hope that you can accept this argumentation. Text was not corrected.
7.
- 90: WHY DID YOU CHOOSE THESE AGE GROUPS? THEY SEEM QUITE RANDOM.
Our intention was to make distinction between young, middle aged and older adults. We choose arbitrary the 40 years and 70 years as thresholds for the categorization.
Original sentence:
The applied age groups were 18-40, 41-70, and 71-105 years.
Corrected sentence (Line 107-108):
The applied age groups were 18-40, 41-70, and 71-105 years, to make distinction between the young, middle aged and older adults.
8.
RESULTS
PLEASE INCLUDE SUBHEADINGS TO STRUCTURE THE RESULTS PART.
Thanks for this notice! Corrected as suggested. The subheadings inserted are:
3.1 Descriptive statistics of the investigated sample
3.2 Associations by univariate analyses
3.3 Associations by multivariate analyses
9.
- 137-143 PLEASE REPORT THE P VALUES IN THE TEXT.
The adjusted odds ratios were applied to describe the association between influencing factors and outcome measures. We used 95% confidence interval based interpretation instead of the statistical test derived p-value based interpretation. The 95% confidence intervals can demonstrate not only the significance of the association but the precision of the estimations as well (by the width of the intervals) The p-value could demonstrate only the significance of the adjusted odds ratios, but not the precision of the estimations. Hopefully, You can accept this approach as appropriate.
10.
- 138 PLEASE DEFINE ELDERIES, WHICH AGE GROUP IS THIS?
The studied subjects of the highest age group (above 70 years of age) were considered as elderlies. The misleading term has been reformulated.
Original sentence:
Elderlies were less infected …
Corrected sentence (Line 158-159):
Persons at least 70 years old were less infected …
11.
THE SAME GOES FOR MORE OR LESS EDUCATED PARTICIPANTS.
The education was categorized as primary, secondary, and tertiary in the investigation. Education showed significant association with each outcome according to the univariate analyses. People with primary level education had lower risk of infection, lower testing frequency, and contact tracing. But for vaccination, the people with primary or secondary education showed lower vaccination rate. So the primary education alone for the first three outcomes, and the primary and secondary level of education for the fourth outcome were applied in the interpretation. The text has been corrected accordingly.
Original sentence:
In the survey, the less educated proved to be less infected, less tested, and less vaccinated.
Corrected sentence (Line 159-160):
In the survey, the primary educated proved to be less infected, less tested, and less vaccinated. Contact tracing was more frequent among tertiary educated persons.
12.
- 140 WHEN YOU GET TESTED LESS, YOU PROBABLY NOTICE LESS INFECTIONS. PLEASE INCLUDE THIS IN THE LIMITATIONS.
It is obvious that all persons with infections were not identified due to the low testing activity. There were subjects with subclinical infection without perception of the infection. They were not tested because the test based, organized screening program was restricted to selected high-risk populations (defined by occupation) in Hungary. Additionally, as it was quantified by the survey, that there were persons with recognized symptoms but without test confirmation. Consequently, persons with infection confirmed by laboratory finding considered as infected in our analyses are smaller group than all the infected persons. This uncertainty was the cause of the restriction of the infection related analysis to the population which has laboratory confirmed diagnosis. This paragraph has been added to the 4.2 Strengths and limitations to admit this limitation (Line 273-280):
All persons with infections were not identified in the survey due to the low testing activity. There were subjects with subclinical infection without perception of the infection. They were not tested because the test based, organized screening program was restricted to selected high-risk populations (defined by occupation) in Hungary. Additionally, as it was directly quantified by the survey, that there were persons with recognized symptoms but there was no test confirmation. Consequently, persons with infection confirmed by laboratory finding, who were considered as infected in our analyses were composed of a subgroup of all infected persons.
13.
- 142 was associated how? Positive or negative?
Thanks for this remark! The sentence has been corrected accordingly.
Original sentence:
A higher BMI was associated with vaccination.
Corrected sentence (Line 164):
The BMI was higher among vaccinated than among not vaccinated.
14.
TABLE 2. WHY IS THE DENOMINATOR OF DISTRIBUTION AND INFECTED NOT THE SAME?
As it is described in the 2.2 Outcome variables the infection is defined as laboratory confirmed infection. There were participants who reported infection on the base of symptoms without lab findings. These subjects were rejected from the analysis because of their uncertain status. The outcome definitions has been added to the footnote of the Table 2:
# infections confirmed by laboratory findings
15.
HOW DID YOU TEST WHEN YOU HAD MORE THAN TWO EXPRESSIONS? E. G. AGE, TO WHICH GROUPS DOES THE P-VALUE REFER?
If there were more than two categories for explanatory variable, then chi-square test was applied as well. In the case of age and education, 2-by-3 tables were used. P values of the Table 2 are the results from these chi square tests. We did not correct the manuscript. Hopefully, you can accept this explanation.
16.
DISCUSSION
PLEASE GIVE A SHORT OVERVIEW ABOUT THE MAIN FINDINGS AT THE BEGINNING OF THE DISCUSSION.
The structure of the 4.1.Main findings is to mention an important observation as a first sentence of a paragraph, followed by the interpretation in the other part of the paragraph. Hopefully, you can accept this structure. Text was not corrected.
17.
- 194 HOW CAN THIS BE CONSISTENT WITH THE RECOMMENDATIONS? IS IT RECOMMEND TO E.G. TEST HIGHLY EDUCATED INDIVIDUALS MORE OFTEN? PLEASE REPHRASE.
The WHO recommendation define occupational high risk groups (e.g.: healthcare staff). Following this recommendation the direct consequence is that the high-risk occupational groups are more tested than the population average, and it is reflected in the higher testing frequency among middle aged, employed and more educated.
Original sentence:
Middle-aged [38,41,42] employed [38,41,42,43] and highly educated individuals [42,44] most frequently received testing, consistent with the WHO recommendations [58] and published international experiences [57].
Corrected sentence (Line 212-215):
Middle-aged [38,41,42] employed [38,41,42,43] and highly educated individuals [42,44] most frequently received testing, consistent with published international experiences [57]. It is the consequence of the adaption of the high-risk target groups’ definition for screening group by the WHO [58].
18.
- 211 WHY 1.5 TIMES LARGER? WHERE DO YOU GET THIS NUMBER FROM?
The first sentence of the paragraph declare that “The coverage of contact tracing among infected individuals was 67.9% in our survey, …” Two-third of the subjects was contact traced, one-third of them was not. This is the source of the “1.5 times larger” interpretation. The interpretation sentence has been reformulated accordingly.
Original sentence:
A contact tracing team 1.5 times larger than the one used would have been beneficial.
Corrected sentence (Line 231-233):
A contact tracing team 1.5 times larger than the one used would have been beneficial, since it could ensure reaching all infected instead of the 67.9% of them.
19.
- 219 PLEASE BE CAREFUL WITH ASSUMPTIONS. YOU CAN ONLY GUESS THAT THE IMPACT MIGHT BE MORE PROFOUND, NOT THAT IT IS.
Thanks for this remark! The overinterpretation has been corrected.
Original sentence:
This impact was more profound among non-middle-aged, unemployed, and primary educated adults and those without chronic disease.
Corrected sentence (Line 240-242):
This impact could be more profound among non-middle-aged, unemployed, and primary educated adults and those without chronic disease.
20.
- 231 How did you observe if the obese population had an adverse immune response from your data?
We did not investigate the immune response. Publications referred at the end of the sentence did. The position of references is wrong. It is misleading. The references have been inserted into the proper place within the sentence.
Original sentence:
It has been demonstrated that being obese is a significant factor in the likelihood of adverse immune responses to SARS-CoV-2 vaccination; however, this observation is contrary to our findings [65,66].
Corrected sentence (Line 253-255):
It has been demonstrated that being obese is a significant factor in the likelihood of adverse immune responses to SARS-CoV-2 vaccination [68,69]; however, this observation is contrary to our findings.
21.
- 241 WHAT TIME PERIOD DID THEY HAVE TO RECALL? AROUND 6 MONTH SINCE THE BEGINNING OF THE PANDEMIC?
The COVID-19 epidemic started in March 2020 in Hungary. The recent recall means recall for 12-14 months. The sentence has been corrected accordingly.
Original sentence:
Although each outcome was assessed by self-reporting, the participants were asked about their recent experiences regarding a very important issue. Corrected sentence (Line 262-263):
Although each outcome was assessed by self-reporting, the participants were asked about their recent, 12 to 14 months experiences regarding a very important issue.
22.
- 259 Is the risk taking behavior the only possible explanation for a lower vaccination rate? What about the interaction with lower educational level etc.?
In this sentence the adjusted odds ratios are interpreted which are corrected by the explanatory variables included in the regression models. On the other hand, the risk taking behavior is only one possible explanation for the observation. To emphasize it, another possible explanation was added to the sentence, and it is emphasized that this is only a possible explanation without supporting results from our analysis.
Original sentence:
Smokers’ risk-taking behavior [70] is reflected in their lower vaccination rate.
Corrected sentence (Line 289-291):
Smokers’ risk-taking behavior [73,74] and smokers’ believe that they are protected against the severe manifestation of COVID-19 infection [75] can be reflected in their lower vaccination rate.
23.
Conclusion
Please refer back to the study aims and if/how you completed them.
To make it explicit, how the study aims were completed: answers to the study questions has been signed by insertion of the number of question applied in the study aims paragraph.
Original sentence:
The socioeconomic or lifestyle-related inequalities in test-confirmed SARS-CoV2 infection were not confirmed in Hungary. However, the survey indicated that testing, contact tracing and vaccination were seriously influenced by socioeconomic position, less so by chronic disease prevalence and very minimally by lifestyle. Considering that the socioeconomic inequalities in COVID-19-related deaths have been demonstrated convincingly in Hungary and that epidemic measures are obviously effective, the etiological role of socioeconomic inequalities in epidemic measure implementation likely generated socioeconomic inequality in COVID-19-related death rates.
Corrected sentence (Line 323-332):
(1) It was the first Hungarian study on the person reported experiences about epidemiological control measures. (2) The socioeconomic or lifestyle-related inequalities in test-confirmed SARS-CoV2 infection were not confirmed in Hungary. However, the survey indicated that testing, contact tracing and vaccination were seriously influenced by socioeconomic position, less so by chronic disease prevalence and very minimally by lifestyle. (3) Considering that the socioeconomic inequalities in COVID-19-related deaths have been demonstrated convincingly in Hungary and that epidemic measures are obviously effective, the etiological role of socioeconomic inequalities in epidemic measure implementation likely generated socioeconomic inequality in COVID-19-related death rates.

Round 2
Reviewer 3 Report
Dear authors,
thank you for the revision. I can accept most of the revisions, except two.
2. I still miss information about how these interviews took place. Were they in person or on the telephone? This can lead to different limitations.
15. The p-value only indicates that there is a significant difference between e. g. primary, secondary and tertiary educated persons, but not whether the difference is between primary and secondary or primary and tertiary educated persons etc. Consider doing a post-hoc analysis to check for that or rephrase the results.
Author Response
Dear Reviewer,
Thank you very much for the careful review of our revised manuscript. Please find enclosed the new version of the paper “Effectiveness of and inequalities in COVID-19 epidemic control strategies in Hungary: A nationwide cross-sectional study” by János Sándor, et al.
Each comment has been considered. The corresponding changes and refinements made in the revised paper are summarized in our response after considering each of your suggestion. Answers along with the modifications we made are summarized below (comments/questions of Yours are in capitals).
Sincerely yours, Janos Sandor (on behalf of the authors)
Answers/reflections to the comments of Reviewer-3:
1.
I STILL MISS INFORMATION ABOUT HOW THESE INTERVIEWS TOOK PLACE. WERE THEY IN PERSON OR ON THE TELEPHONE? THIS CAN LEAD TO DIFFERENT LIMITATIONS.
The interviews were carried out in person. The text has been corrected accordingly in two sentences (in the abstract and in the methods section).
Original sentence in the Abstract:
A nationwide representative sample of 1008, randomly selected adults were interviewed in person between March 15 and May 30, 2021.
Corrected sentence (Line 17-19):
A nationwide representative sample of 1008, randomly selected adults were interviewed in person between March 15 and May 30, 2021.
Original sentence in the Materials and Methods:
A representative sample of 1008 Hungarian adults over 18 years of age was randomly selected from the country’s whole population (using the national registry of the Hungarian population as sampling frame) and interviewed by trained interviewers [17].
Corrected sentence (Line 87-90):
A representative sample of 1008 Hungarian adults over 18 years of age was randomly selected from the country’s whole population (using the national registry of the Hungarian population as sampling frame) and interviewed in person by trained interviewers [17].
2.
THE P-VALUE ONLY INDICATES THAT THERE IS A SIGNIFICANT DIFFERENCE BETWEEN E. G. PRIMARY, SECONDARY AND TERTIARY EDUCATED PERSONS, BUT NOT WHETHER THE DIFFERENCE IS BETWEEN PRIMARY AND SECONDARY OR PRIMARY AND TERTIARY EDUCATED PERSONS ETC. CONSIDER DOING A POST-HOC ANALYSIS TO CHECK FOR THAT OR REPHRASE THE RESULTS.
2.a
The post-hoc analysis was not carried out because the univariate analysis did not determine the conclusion. The results of the post-hoc analysis are summarized in the Appendix. The additional analysis led to some modification of sentences, because there were a few post-hoc test results above the Bonferroni corrected threshold (pthreshold=0.05/6=0.008)
Original sentences:
Persons at least 70 years old were less infected and less tested, but more contact tracing and more vaccination occurred. In the survey, the primary educated proved to be less infected, less tested, and less vaccinated. Contact tracing was more frequent among tertiary educated persons.
Corrected sentences: (Line 156-159):
Persons at least 70 years old were less infected and less tested, but more vaccination occurred. In the survey, the primary educated proved to be less infected, and less test-ed. While the tertiary educated were more tested and vaccinated.
2.b
The inserted sentence (Line 162-163):
(Results of the post-hoc analysis are summarized in the Appendix.)
2.c
The Appendix is the following:
- Post-hoc analysis for the chi-square test on the association between age and test confirmed SARS-CoV2 infection
|
age |
not infected |
infected |
total |
|
|
middle |
Observed |
508 |
82 |
590 |
|
Expected |
518.503 |
71.497 |
590 |
|
|
Adjusted Residual |
-2.097 |
2.097 |
||
|
P-value |
0.036 |
0.036 |
||
|
older |
Observed |
125 |
5 |
130 |
|
Expected |
114.246 |
15.754 |
130 |
|
|
Adjusted Residual |
3.103 |
-3.103 |
||
|
P-value |
0.002 |
0.002 |
||
|
young |
Observed |
230 |
32 |
262 |
|
Expected |
230.251 |
31.749 |
262 |
|
|
Adjusted Residual |
-0.055 |
0.055 |
||
|
P-value |
0.956 |
0.956 |
||
|
total |
Observed |
863 |
119 |
982 |
|
Expected |
863 |
119 |
982 |
- Post-hoc analysis for the chi-square test on the association between age and testing for SARS-CoV2 infection
|
age |
not tested |
tested |
total |
|
|
middle |
Observed |
368 |
238 |
606 |
|
Expected |
401.595 |
204.405 |
606 |
|
|
Adjusted Residual |
-4.571 |
4.571 |
||
|
P-value |
<0.001 |
<0.001 |
||
|
older |
Observed |
115 |
17 |
132 |
|
Expected |
87.476 |
44.524 |
132 |
|
|
Adjusted Residual |
5.435 |
-5.435 |
||
|
P-value |
<0.001 |
<0.001 |
||
|
young |
Observed |
185 |
85 |
270 |
|
Expected |
178.929 |
91.071 |
270 |
|
|
Adjusted Residual |
0.913 |
-0.913 |
||
|
P-value |
0.361 |
0.361 |
||
|
total |
Observed |
668 |
340 |
1008 |
|
Expected |
668 |
340 |
1008 |
- Post-hoc analysis for the chi-square test on the association between age and contact tracing
|
age |
not contact traced |
contact traced |
total |
|
|
middle |
Observed |
23 |
67 |
90 |
|
Expected |
29.318 |
60.682 |
90 |
|
|
Adjusted Residual |
-2.519 |
2.519 |
||
|
P-value |
0.012 |
0.012 |
||
|
older |
Observed |
3 |
4 |
7 |
|
Expected |
2.28 |
4.72 |
7 |
|
|
Adjusted Residual |
0.596 |
-0.596 |
||
|
P-value |
0.551 |
0.551 |
||
|
young |
Observed |
17 |
18 |
35 |
|
Expected |
11.402 |
23.598 |
35 |
|
|
Adjusted Residual |
2.355 |
-2.355 |
||
|
P-value |
0.018 |
0.018 |
||
|
total |
Observed |
43 |
89 |
132 |
|
Expected |
43 |
89 |
132 |
- Post-hoc analysis for the chi-square test on the association between age and vaccination against COVID-19
|
age |
not vaccinated |
vaccinated |
total |
|
|
middle |
Observed |
338 |
268 |
606 |
|
Expected |
317.429 |
288.571 |
606 |
|
|
Adjusted Residual |
2.65 |
-2.65 |
||
|
P-value |
0.008 |
0.008 |
||
|
older |
Observed |
113 |
19 |
132 |
|
Expected |
69.143 |
62.857 |
132 |
|
|
Adjusted Residual |
8.199 |
-8.199 |
||
|
P-value |
<0.001 |
<0.001 |
||
|
young |
Observed |
77 |
193 |
270 |
|
Expected |
141.429 |
128.571 |
270 |
|
|
Adjusted Residual |
-9.175 |
9.175 |
||
|
P-value |
<0.001 |
<0.001 |
||
|
total |
Observed |
528 |
480 |
1008 |
|
Expected |
528 |
480 |
1008 |
- Post-hoc analysis for the chi-square test on the association between level of infection and test confirmed SARS-CoV2 infection
|
education |
not infected |
infected |
total |
|
|
primary |
Observed |
152 |
9 |
161 |
|
Expected |
141.49 |
19.51 |
161 |
|
|
Adjusted Residual |
2.776 |
-2.776 |
||
|
P-value |
0.006 |
0.006 |
||
|
secondary |
Observed |
585 |
90 |
675 |
|
Expected |
593.203 |
81.797 |
675 |
|
|
Adjusted Residual |
-1.73 |
1.73 |
||
|
P-value |
0.084 |
0.084 |
||
|
tertiary |
Observed |
126 |
20 |
146 |
|
Expected |
128.308 |
17.692 |
146 |
|
|
Adjusted Residual |
-0.634 |
0.634 |
||
|
P-value |
0.526 |
0.526 |
||
|
total |
Observed |
863 |
119 |
982 |
|
Expected |
863 |
119 |
982 |
- Post-hoc analysis for the chi-square test on the association between level of education and testing for SARS-CoV2 infection
|
education |
not tested |
tested |
total |
|
|
primary |
Observed |
139 |
26 |
165 |
|
Expected |
109.345 |
55.655 |
165 |
|
|
Adjusted Residual |
5.34 |
-5.34 |
||
|
P-value |
<0.001 |
<0.001 |
||
|
secondary |
Observed |
453 |
241 |
694 |
|
Expected |
459.913 |
234.087 |
694 |
|
|
Adjusted Residual |
-0.994 |
0.994 |
||
|
P-value |
0.320 |
0.320 |
||
|
tertiary |
Observed |
76 |
73 |
149 |
|
Expected |
98.742 |
50.258 |
149 |
|
|
Adjusted Residual |
-4.269 |
4.269 |
||
|
P-value |
<0.001 |
<0.001 |
||
|
total |
Observed |
668 |
340 |
1008 |
|
Expected |
668 |
340 |
1008 |
- Post-hoc analysis for the chi-square test on the association between level of education and contact tracing
|
education |
not contact traced |
contact traced |
total |
|
|
primary |
Observed |
6 |
6 |
12 |
|
Expected |
3.909 |
8.091 |
12 |
|
|
Adjusted Residual |
1.351 |
-1.351 |
||
|
P-value |
0.177 |
0.177 |
||
|
secondary |
Observed |
32 |
68 |
100 |
|
Expected |
32.576 |
67.424 |
100 |
|
|
Adjusted Residual |
-0.25 |
0.25 |
||
|
P-value |
0.803 |
0.803 |
||
|
tertiary |
Observed |
5 |
15 |
20 |
|
Expected |
6.515 |
13.485 |
20 |
|
|
Adjusted Residual |
-0.785 |
0.785 |
||
|
P-value |
0.433 |
0.433 |
||
|
total |
Observed |
43 |
89 |
132 |
|
Expected |
43 |
89 |
132 |
- Post-hoc analysis for the chi-square test on the association between level of education and vaccination against COVID-19
|
education |
not vaccinated |
vaccinated |
total |
|
|
primary |
Observed |
87 |
78 |
165 |
|
Expected |
86.429 |
78.571 |
165 |
|
|
Adjusted Residual |
0.097 |
-0.097 |
||
|
P-value |
0.922 |
0.922 |
||
|
secondary |
Observed |
337 |
357 |
694 |
|
Expected |
363.524 |
330.476 |
694 |
|
|
Adjusted Residual |
-3.612 |
3.612 |
||
|
P-value |
<0.001 |
<0.001 |
||
|
tertiary |
Observed |
104 |
45 |
149 |
|
Expected |
78.048 |
70.952 |
149 |
|
|
Adjusted Residual |
4.611 |
-4.611 |
||
|
P-value |
<0.001 |
<0.0010 |
||
|
total |
Observed |
528 |
480 |
1008 |
|
Expected |
528 |
480 |
1008 |
